# Identification and Characterization of RK22, a Novel Antimicrobial Peptide from *Hirudinaria manillensis* against Methicillin Resistant *Staphylococcus aureus*

**DOI:** 10.3390/ijms241713453

**Published:** 2023-08-30

**Authors:** Xiaoyu Lu, Min Yang, Shengwen Zhou, Shuo Yang, Xiran Chen, Mehwish Khalid, Kexin Wang, Yaqun Fang, Chaoming Wang, Ren Lai, Zilei Duan

**Affiliations:** 1School of Life Sciences, Tianjin University, Tianjin 300072, China; o0o0o00823@163.com; 2Key Laboratory of Bioactive Peptides of Yunnan Province, National & Local Joint Engineering Center of Natural Bioactive Peptides, Kunming Institute of Zoology, Chinese Academy of Sciences, Kunming 650223, China; yangmin@mail.kiz.ac.cn (M.Y.); zhoushengwen@mail.kiz.ac.cn (S.Z.); yangshuo@mail.kiz.ac.cn (S.Y.); chenxiran@mail.kiz.ac.cn (X.C.); mehwishkhalidd120@gmail.com (M.K.); kx_1998@tju.edu.cn (K.W.); fangyaqun@mail.kiz.ac.cn (Y.F.); wangchaoming@mail.kiz.ac.cn (C.W.); 3University of Chinese Academy of Sciences, Beijing 101408, China; 4School of Medical Engineering and Translational Medicine, Tianjin University, Tianjin 300072, China; 5KIZ-CUHK Joint Laboratory of Bioresources and Molecular Research in Common Diseases, Kunming Institute of Zoology, Chinese Academy of Sciences, Kunming 650107, China; 6National Resource for Non-Human Primates, Kunming Primate Research Center, National Research Facility for Phenotypic & Genetic Analysis of Model Animals (Primate Facility), Kunming Institute of Zoology, Chinese Academy of Sciences, Kunming 650107, China; 7Sino-African Joint Research Center, Kunming Institute of Zoology, Chinese Academy of Sciences, Kunming 650223, China

**Keywords:** *Staphylococcus aureus*, antimicrobial peptide, RK22, *Hirudinaria manillensis*

## Abstract

*Staphylococcus aureus* (*S. aureus*) infections are a leading cause of morbidity and mortality, which are compounded by drug resistance. By manipulating the coagulation system, *S. aureus* gains a significant advantage over host defense mechanisms, with hypercoagulation induced by *S. aureus* potentially aggravating infectious diseases. Recently, we and other researchers identified that a higher level of LL-37, one endogenous antimicrobial peptide with a significant killing effect on *S. aureus* infection, resulted in thrombosis formation through the induction of platelet activation and potentiation of the coagulation factor enzymatic activity. In the current study, we identified a novel antimicrobial peptide (RK22) from the salivary gland transcriptome of *Hirudinaria manillensis* (*H. manillensis*) through bioinformatic analysis, and then synthesized it, which exhibited good antimicrobial activity against *S. aureus*, including a clinically resistant strain with a minimal inhibitory concentration (MIC) of 6.25 μg/mL. The RK22 peptide rapidly killed *S. aureus* by inhibiting biofilm formation and promoting biofilm eradication, with good plasma stability, negligible cytotoxicity, minimal hemolytic activity, and no significant promotion of the coagulation system. Notably, administration of RK22 significantly inhibited *S. aureus* infection and the clinically resistant strain in vivo. Thus, these findings highlight the potential of RK22 as an ideal treatment candidate against *S. aureus* infection.

## 1. Introduction

*Staphylococcus aureus* (*S. aureus*), a common colonizer of the human population, is recognized as a significant opportunistic bacterial pathogen responsible for a considerable burden of morbidity and mortality [1]. This bacterium can colonize various body sites and organs, posing a risk for the development of infections, ranging from mild skin and soft tissue infections to severe invasive diseases [2,3]. While the introduction of penicillin and methicillin initially exhibited favorable therapeutic efficacy against *S. aureus*, their extensive use has ultimately led to the emergence of antibiotic resistance [4]. Despite the mitigating effects of vancomycin, the subsequent identification of vancomycin-resistant *S. aureus* has necessitated further research into novel antibacterial agents [5,6] to ensure that effective treatment options for emerging drug-resistant *S. aureus* infections do not become scarce.

Antimicrobial peptides offer a promising alternative to traditional antibiotics, displaying several advantages, such as slower emergence of drug resistance, broad-spectrum antibiofilm activity, and the ability to modulate the host immune response [7]. Among these peptides, defensins and cathelicidins represent two major families of endogenous antimicrobial peptides found in humans [8], which demonstrate a notable bacteriostatic or bactericidal effects on *S. aureus* [9]. However, in addition to their antimicrobial effects, elevated levels of these endogenous antimicrobial peptides are also associated with the pathogenesis of various diseases, including psoriasis [10], atherosclerosis [11,12], colitis [13], Alzheimer’s disease [14], and viral infections [15,16], thereby limiting their applicability in microbial infections related to these diseases. Furthermore, *S. aureus* infections can manipulate and activate the human hemostatic system, gaining an advantage over host defense mechanisms [17,18,19]. Thus, the induction and promotion of coagulation and platelet activation by defensins [12,20,21,22] and LL-37 [16,23,24], the only human member of the cathelicidin antimicrobial peptide family, may aggravate these infections. As such, there is an urgent need for the development of novel antimicrobial peptides that exhibit minimal potentiation of the human hemostatic system.

The salivary glands of hematophagous leeches secrete an abundance of anticoagulant substances [25,26,27,28,29], and recently several novel antimicrobial peptides have been identified from these leeches such as *Hirudo nipponica Whitman* and *Hirudo medicinalis* [30,31,32]. *H. manillensis* is the most commonly available leech from the Chinese commercial leech market, and we and other researchers have identified a variety of anticoagulant peptides, analgesic peptides, and other active molecules from the salivary gland of *H. manillensis* [33,34,35,36,37]. However, no antimicrobial peptides from *H. manillensis* have been reported. In the current study, we screened several positively charged peptides and identified a novel antimicrobial peptide (RK22) from the transcriptome of the salivary glands of *H. manillensis* through bioinformatic analysis, and then determined the effects of synthesized RK22 against *S. aureus*. Results showed that RK22 has good antimicrobial activity against *S. aureus*, which suggested the potential of RK22 as an ideal treatment candidate against *S. aureus* infection.

## 2. Results

### 2.1. Identification and Characterization of Antimicrobial Peptides from the Salivary Gland Transcriptome of H. manillensis

Antimicrobial peptides, predominantly cationic in nature, were discovered in abundance in the transcriptome of the salivary glands of *H. manillensis* (Table 1). To assess their potential as antimicrobial peptides, we conducted a bioinformatic prediction analysis of these peptides using the Antimicrobial Peptide Scanner v2. As illustrated in Table 2, VF-21 and RK22 were predicted as antimicrobial peptides based on random forest class and earth class predictions, while another four were suggested as antimicrobial peptides based on only one prediction. Subsequently, we synthesized these six peptides and determined their MICs against the standard strain. As shown in Table 3, RK22 exhibited potent antibacterial activity against *S. aureus* ATCC6538, *A. baumannii* ATCC19606, and *E. coli* ATCC8739, with MIC values of 6.25, 100, and 25 μg/mL, respectively. However, it showed no antibacterial activity against *P. aeruginosa* ATCC27853, even at concentrations exceeding 100 μg/mL. Among the other peptides, KS14, KT20, and RA19 exhibited antimicrobial activity against *S. aureus* ATCC6538, with MIC values of 50, 50, and 12.5 μg/mL, respectively, but showed no antibacterial activity against *A. baumannii* ATCC19606, *P. aeruginosa* ATCC27853, or *E. coli* ATCC8739, even at concentrations above 100 μg/mL. KS34 and VF21 showed no antibacterial activity against any of the four bacterial strains. Thus, we identified four antimicrobial peptides from the salivary gland transcriptome of *H. manillensis*, among which RK22 was the most promising candidate for further development.

### 2.2. RK22 Exerts Antimicrobial Activity against Clinically Isolated S. aureus and Methicillin-Resistant S. aureus Strain

To further evaluate the antimicrobial activity of RK22 against *S. aureus*, two clinically isolated strains (SA220823 and SA15772), and three clinically isolated methicillin-resistant strains (MRSA-Z, MRSA11, and MRSA22) were selected. As depicted in Table 4, RK22 exhibited potent antibacterial activity against the three clinically isolated methicillin-resistant *S. aureus* strains with MIC values of 6.25 μg/mL and two clinically isolated *S. aureus* strains with MIC values of 12.5 μg/mL. Although slightly weaker than vancomycin (MIC of 1.56 μg/mL), RK22 displayed potent antibacterial activity against both standard and antibiotic-resistant strains of *S. aureus*.

### 2.3. RK22 Exhibits Rapid Killing Ability against S. aureus ATCC6538 and MRSA-Z by Cell Membrane Permeabilization

To explore the potential mechanism underlying the antimicrobial activity of RK22 against *S. aureus*, we employed SEM and TEM to examine the morphology of *S. aureus* ATCC6538 and MRSA-Z. Results showed that RK22 treatment led to an obvious disruption of the plasma membrane in both *S. aureus* ATCC6538 (Figure 1A) and MRSA-Z (Figure 1B), whereas the plasma membrane in the untreated groups remained intact.

Furthermore, to assess the bactericidal kinetics of RK22 against *S. aureus* ATCC6538 and MRSA-Z, we measured its rate of bactericidal activity. Results demonstrated that RK22 administration exerted more rapid killing effects against *S. aureus* ATCC6538 (Figure 1C) compared to MRSA-Z (Figure 1D), even surpassing the speed of vancomycin at the equivalent MIC. Notably, RK22 achieved a kill rate of more than 90% against *S. aureus* ATCC6538at 1 × MIC and 5 × MIC within 3 h and at 10 × MIC within 1 h, whereas vancomycin required 6 h to achieve a kill rate of more than 90% against *S. aureus* ATCC6538 at 5 × MIC and 10 × MIC (Figure 1C). Both RK22 and vancomycin exhibited slower bactericidal activity against MRSA-Z compared to the standard *S. aureus* strain ATCC6538. However, RK22 demonstrated the ability to eradicate almost all MRSA-Z at 10 × MIC within 6 h, while vancomycin only inhibited about 80% of MRSA-Z at the same MIC and time point (Figure 1D). Thus, these findings provide evidence that RK22 exhibits more rapid antibacterial activity against both *S. aureus* ATCC6538 and MRSA-Z in comparison to vancomycin.

### 2.4. RK22 Exhibits Biofilm Inhibition and Eradication Activities

To further investigate the potential mechanism underlying the antimicrobial activity of RK22, we investigated its effects on *S. aureus* ATCC6538 and MRSA-Z biofilm formation and eradication. Results showed that RK22 exhibited a dose-dependent inhibition of biofilm formation in both *S. aureus* ATCC6538 (Figure 2A) and MRSA-Z (Figure 2B). Furthermore, RK22 demonstrated the ability to effectively eliminate preformed biofilms of *S. aureus* ATCC6538 (Figure 2C) and MRSA-Z (Figure 2D) in a dose-dependent manner.

### 2.5. RK22 Maintains Antibacterial Activity in Plasma

The antimicrobial function of antimicrobial peptides can be neutralized by various plasma proteins, while degradation by specific plasma proteases can diminish their antimicrobial efficacy [38]. Here, we investigated the effects of plasma on the antibacterial activity of RK22. As shown in Appendix A, the antimicrobial activity of RK22 against *S. aureus* ATCC6538 and MRSA-Z remained consistent at 6.25 μg/mL, even after 10 h of in vitro plasma incubation. These findings suggest that incubation of RK22 in plasma did not significantly affect its antibacterial activity against *S. aureus* ATCC6538 and MRSA-Z.

### 2.6. RK22 Shows Negligible Hemolytic Activity and Cytotoxicity in Mammalian Cells

The potent hemolytic activity and cytotoxicity exhibited by certain highly effective antimicrobial peptides have hindered their application in vivo. To assess the hemolytic activity of RK22 in blood cells, we performed hemolysis assays. As illustrated in Appendix A, RK22 did not exhibit any obvious hemolytic activity, even at a concentration of 400 μg/mL. Additionally, we evaluated the cytotoxicity of RK22 against HEK293T cells. Results indicated that RK22 did not demonstrate any noticeable cytotoxic effects on HEK293T cells, even at a concentration of 400 μg/mL (Appendix A). Thus, these findings suggest that treatment with RK22 is unlikely to induce hemolytic activity or cytotoxicity in mammalian cells in vivo.

### 2.7. RK22 Induces No Potentiation of Coagulation Factor Activity and Platelet Activation

Upon infection, *S. aureus* bacteria can manipulate and activate the human hemostatic system, providing them an advantage over host defense mechanisms [17,18,19]. It has recently been confirmed that LL-37 can induce and promote coagulation and platelet activation [16,23,24]. Thus, we compared the effects of LL-37 and RK22 on coagulation factor activity and platelet activation. Results showed that while LL-37 demonstrated significant potentiation activity, RK22 did not exhibit similar enhancement of the enzymatic activity of thrombin (Figure 3A,B) or FXa (Figure 3C,D) in the same molar concentration of LL-37. Similarly, while LL-37 exhibited significant and dose-dependent effects on platelet activation, RK22 did not exert any such effects (Figure 3E,F). Thus, these findings suggest that RK22 is unlikely to induce hypercoagulation similar to the effects induced by LL-37.

### 2.8. RK22 Shows No Acute Toxicity In Vivo

To determine the acute toxicity of RK22 in vivo, male C57BL/6 mice (6–8 weeks old) were injected with a single dose of RK22 at concentrations of 20 and 40 mg/kg. Mouse survival was then monitored for a period of 96 h. As shown in Appendix A, treatment with RK22 at both 20 and 40 mg/kg did not result in any fatalities, suggesting low toxicity. The mice were sacrificed after 96 h of RK22 treatment, and blood and tissue samples were collected for further analysis. Results showed that treatment with RK22 did not induce any changes in white blood cells (Appendix A), red blood cells (Appendix A), hemoglobin (HGB) (Appendix A), or platelets (Appendix A). Furthermore, histopathological analysis with hematoxylin and eosin (H&E) staining suggested no obvious tissue injury after stimulation with RK22 (Appendix A).

### 2.9. RK22 Suppresses S. aureus Dissemination

To evaluate the therapeutic potential of RK22 as a candidate drug for *S. aureus* infection, we determined the suppression of *S. aureus* by RK22 in vivo. As shown in Figure 4A,B, treatment with RK22 following *S. aureus* ATCC6538 infection significantly inhibited the dissemination of standard *S. aureus* ATCC6538 from the peritoneal cavity to the blood (Figure 4A) and lung (Figure 4B), similar to the effects of vancomycin. Additionally, histopathological examination revealed that treatment with RK22 (8 mg/kg) and vancomycin (4 mg/kg) significantly alleviated *S. aureus* ATCC6538 infection-induced hemorrhage and inflammatory cell infiltration in the lung (Figure 4C). Furthermore, RK22 significantly suppressed the dissemination of MRSA-Z from the peritoneal cavity to the blood (Figure 4D) and lung (Figure 4E). Consistently, treatment with RK22 (8 mg/kg) and vancomycin (4 mg/kg) significantly alleviated MRSA-Z infection-induced tissue injury in the lung (Figure 4F). These findings confirm the ability of RK22 to suppress the dissemination of *S. aureus* ATCC6538 and MRSA-Z to target organs in vivo.

## 3. Discussion

*S. aureus* and MRSA infections represent a significant threat to human health, emphasizing the critical need for the development of novel antimicrobial drugs. In the present study, we identified a novel antimicrobial peptide (RK22), which selectively inhibited infection of *S. aureus* and clinically resistant *S. aureus* strains in vitro and in vivo. Notably, RK22 exhibited rapid bactericidal activity against *S. aureus* by inhibiting biofilm formation and promoting biofilm eradication, while simultaneously showing good plasma stability, negligible cytotoxicity, minimal hemolytic activity, and no significant promotion of the coagulation system.

Hematophagous leeches have garnered increasing attention as potential sources of novel antimicrobial molecules due to their ability to store ingested blood for extended periods without significant changes. Indeed, previous studies have identified several antimicrobial peptides derived from such leeches [30,31,32,39,40]. Vakhrusheva et al. discovered that certain leech-derived antimicrobial peptides can prolong activated partial thromboplastin time (APTT) and prothrombin time (PT), indicating their inhibitory effects on coagulation [40]. However, these peptides also displayed obvious hemolysis, which could be mitigated by the presence of blood plasma and albumin [40]. In contrast, our findings demonstrated that RK22 exhibits excellent antimicrobial activity without potentiating coagulation factor enzymatic activity or platelet activation.

Endogenous antimicrobial peptides, such as LL-37, have demonstrated significant bacteriostatic or bactericidal effects on *S. aureus* through direct bacterial killing or regulation of the immune system [41]. However, although LL-37 exhibits good antibacterial, antifungal, and antiviral activity, research has also indicated that higher levels of LL-37 may contribute to the pathogenesis of various diseases. Notably, recent studies have revealed that LL-37 can induce hypercoagulation by potentiating the enzymatic activities of coagulation factors [16] and promoting platelet activation [23,24], which may exacerbate *S. aureus* infections [42], thus highlighting the need for further research. In the current study, we compared the effects of RK22 and LL-37 on coagulation factor activity and platelet activation. Our results showed that RK22 did not enhance the enzymatic activity of thrombin or FXa or induce platelet activation, unlike LL-37, which exhibited significant hypercoagulation effects at the same concentrations as RK22 (Figure 3).

Results of SEM and TEM suggested the direct bacterial killing effects of RK22 (Figure 1A,B), and experiments of the biofilm inhibition assay and biofilm eradication assay suggested the mechanisms of RK22 against *S. aureus* (Figure 2). RK22 showed the most excellent activity with a +10 net charge and VF21 showed no antibacterial activity with a +1 net charge (Table 1 and Table 3), which suggested the more net positive charge, the better antimicrobial activity. However, KS14, RA19, KT20, and KS34 with the same net charge (+7) showed different antimicrobial activity (Table 1 and Table 3). Furthermore, almost all the positively charged peptides (KS14, RA19, KT20, and RK22) showed excellent antimicrobial activity against *S. aureus*, which suggested that positively charged peptides are prone to kill *S. aureus*. Meanwhile, KS34 and VF21 were the exceptions, which suggests that the charge and structure of antimicrobial peptides might determine the activity and the selectivity. More experiments on the activity and selectivity of antimicrobial peptides need to be investigated in the future.

In addition to endogenous antimicrobial peptides, researchers have identified thousands of antimicrobial peptides, leading to the establishment of several antimicrobial peptide databases. These databases provide valuable bioinformatics support for the identification and design of novel antimicrobial peptides, including antimicrobial peptides against *S. aureus* [43], *Candida albicans* [44], *P. aeruginosa*, and *A. baumannii* [45], as reported in our previous research. In the present study, RK22 exhibited antimicrobial activity against *S. aureus*, *A. baumannii*, and *E. coli*, with MIC values of 6.25, 100, and 25 μg/mL, respectively (Table 3). In the future, we aim to utilize RK22 as a foundation for developing more active and specific antimicrobial peptides through amino acid substitutions and modifications. Furthermore, we anticipate that further bioinformatics analyses of the salivary gland transcriptome of *H. manillensis* will uncover additional potent antimicrobial peptides.

While antimicrobial efficacy is a crucial factor determining the applicability of antimicrobial peptides, additional properties, such as plasma stability, hemolytic activity, and cytotoxicity, can significantly influence their potential use [46]. Remarkably, RK22 maintained its antibacterial activity against both *S. aureus* and MRSA, even after 10 h of incubation in plasma (Appendix A). Furthermore, RK22 showed no hemolytic activity and cytotoxicity even at the concentration of 400 μg/mL (Appendix A), which may explain its lack of acute toxicity even at the high concentration of 40 mg/kg (Appendix A).

## 4. Materials and Methods

### 4.1. Bacterial Strain Preparation and Growth Conditions

The bacterial strains used in this study included *Pseudomonas aeruginosa* (*P. aeruginosa*, ATCC27853), *Acinetobacter baumannii* (*A. baumannii*, ATCC19606), *Escherichia coli* (*E. coli*, ATCC8739), and *S. aureus* (ATCC6538, two clinically isolated strains (SA220823 and SA15772)), and three clinically isolated methicillin-resistant strains (MRSA-Z, MRSA11, and MRSA22). All strains were grown in Luria-Bertani (LB) broth.

### 4.2. Animals

C57BL/6 male mice (20–22 g, 6–8 weeks of age) were obtained from SPF (Beijing, China) Biotechnology Co., Ltd. (Beijing, China) for the in vivo experiments, and were maintained on a 12-h light/12-h dark cycle at 24 °C. All experiments were approved by the Institutional Review Board and Animal Care and Use Committee at the Kunming Institute of Zoology, China (IACUC-RE-2023-05-009).

### 4.3. Antibacterial Peptide Identification, Prediction, and Synthesis

Using the assembled high-quality transcriptome data obtained from the salivary gland of *H. manillensis*, we predicted and analyzed positively charged amino acid-rich peptides using the Antimicrobial Peptide Scanner vr.2 (https://www.dveltri.com/ascan/v2/index.html) and ExPASy Swiss Bioinformatics Resource Portal (https://www.expasy.org/), respectively. All peptides used were synthesized by GL Biochem (Shanghai, China) to a purity of more than 95%, as confirmed through reverse-phase high-performance liquid chromatography (RP-HPLC) and mass spectrometry.

### 4.4. In Vitro Antimicrobial Testing

The minimum inhibitory concentration (MIC) for all strains was determined using the tube microdilution assay based on a previous study with minor modifications [47]. Initially, strains in the exponential phase were diluted to a concentration of 2 × 10^5^ CFU/mL using RPMI 1640 medium (Corning, New York, NY, USA). Next, 100 μL of the bacterial suspension was mixed with 100 μL of the sample solution (final concentration 0–200 μg/mL in RPMI 1640 medium) in 96-well plates. The plates were then incubated at 37 °C for 12–16 h. After incubation, the absorbance at 600 nm was measured, and the MIC was determined as the lowest concentration of the test sample in which no bacterial growth was detected in the well.

### 4.5. Hemolysis and Cytotoxicity Assays

The hemolysis and cytotoxicity assays were performed according to previous research with some modifications [48]. In the hemolysis assay, mouse blood was collected in anticoagulant tubes, centrifuged (1000× *g*, 5 min), and washed with normal saline three times to obtain red blood cells. The red blood cells were then resuspended to prepare a 10% red blood cell suspension. The suspension was mixed with an equal volume solution of RK22 (final concentration 0–400 μg/mL), then incubated at 37 °C for 1 h and centrifuged (1000× *g*, 10 min). After this, 150 μL of the supernatant was transferred to a new 96-well plate, with absorbance then measured at 540 nm. Additionally, 1% Triton X-100 and sterile saline treatment groups accounted for 100% and 0% hemolysis, respectively.

In the cytotoxicity assay, HEK293T cells were cultured with Dulbecco’s Modified Eagle Medium (DMEM) (Corning, New York, NY, USA) consisting of 10% fetal bovine serum (FBS, Corning, New York, NY, USA) and 1% penicillin-streptomycin liquid (Solarbio, Beijing, China). After digestion using trypsin (Gibco, New York, NY, USA), the cell suspension (100 μL, 1 × 10^5^ cells/mL, 200 µL/well) was inoculated into each well of a 96-well plate. Once the cells were completely adherent, the supernatant was discarded, and fresh medium with or without 100 μL of RK22 (final concentration 6.25–400 μg/mL) was added to the 96-well plates. After incubation at 37 °C for 24 h, 10 μL of CCK8 (MedChemExpress, South Brunswick Township, NJ, USA) solution was added to each well. The plate was then incubated at 37 °C for an additional 1–4 h, with absorbance then measured at 450 nm.

### 4.6. Changes in Antimicrobial Activity of RK22 in Plasma

We determined the stability of RK22 in plasma according to the previously reported method [45]. Briefly, an equal volume of RK22 was mixed with human plasma and diluted to a concentration of 5 mg/mL. The MICs of RK22 against *S. aureus* ATCC6538 and methicillin-resistant *S. aureus* MRSA-Z were determined after incubation at 37 °C for different times (0, 2, 4, 6, 8, and 10 h).

### 4.7. Evaluation of Bacterial Membrane Morphology

Bacterial membrane morphology was observed using transmission electron microscopy (TEM) and scanning electron microscopy (SEM) following previous research with minor modifications [49]. First, *S. aureus* ATCC6538 or methicillin-resistant *S. aureus* MRSA-Z was cultured in LB broth until reaching the exponential growth phase. The bacterial solution was then washed three times and diluted to 4 × 10^7^ CFU/mL, then treated with RK22 (final concentration 100 μg/mL) or phosphate-buffered saline (PBS, Corning, New York, NY, USA) and incubated at 37 °C for 4 h. The bacterial liquid was centrifuged (400× *g*, 10 min, 4 °C), the remaining bacterial pellets were fixed with 2.5% glutaraldehyde (Servicebio, Wuhan, China) at 4 °C for 4 h, and then post-fixed with 1% buffered osmium tetroxide and 1.5% potassium ferricyanide for 2 h. After fixation, dehydration, embedding, and staining processes, membrane morphology was observed using SEM (JEOL, JEM-1400 Plus, Tokyo, Japan) and TEM (Hitachi, SU-8100, Tokyo, Japan).

### 4.8. Bacterial Killing Kinetic Assay

The bactericidal killing kinetic assay was performed according to a previous study with minor modifications [50]. Firstly, *S. aureus* ATCC6538 or methicillin-resistant *S. aureus* MRSA-Z was diluted to 2 × 10^5^ CFU/mL using RPMI 1640 medium. Equal volumes of either RK22 (final concentration 1, 5, 10 × MIC) or vancomycin (Macklin, Shanghai, China) (final concentration 1, 5, 10 × MIC) solutions were added to the bacterial suspension. The mixtures were then incubated at 37 °C for different times (0, 15, 30, 60, 180, and 360 min). At each time point, the bacterial solution was inoculated on LB agar culture medium, and colony forming units (CFUs) were counted after 24 h.

### 4.9. Biofilm Inhibition Assay

To determine the ability of RK22 to inhibit biofilms, a biofilm inhibition assay was performed based on previous research with minor modifications [45]. In brief, *S. aureus* ATCC6538 or methicillin-resistant *S. aureus* MRSA-Z was diluted to a concentration of 2 × 10^6^ CFU/mL using RPMI 1640 medium. Subsequently, 100 μL of the bacterial suspension and 100 μL of RK22 (final concentration 0.5, 1, 2, 4, 8 × MIC) were mixed in 96-well plates. The plates were then incubated at 37 °C for 24 h, then washed three times with PBS, and fixed with 99% methanol (Damao Chemical Reagent Factory, Tianjin, China) for 15 min. After drying, the methanol was aspirated, and the biofilms were stained with 100 μL of 0.1% crystal violet (Solarbio, Beijing, China) for 5 min. The plates were thrice washed with sterile normal saline solution, and the stain was resolubilized in 95% ethanol (Damao Chemical Reagent Factory, Tianjin, China). Finally, absorbance was measured at 600 nm after 30 min.

### 4.10. Biofilm Eradication Assay

*S. aureus* ATCC6538 or methicillin-resistant *S. aureus* MRSA-Z (2 × 10^6^ CFU/mL) was grown in 100 μL of RPMI 1640 medium in 96-well plates, followed by incubation at 37 °C for 24 h to generate biofilms. After washing three times with PBS, 100 μL of PBS or RK22 (final concentration 0.5, 1, 2, 4, 8 × MIC) was added to each well. The plates were incubated at 37 °C for 24 h, then washed three times with PBS, and fixed with 99% methanol for 15 min. After drying the well, the methanol was aspirated and 100 μL of 0.1% crystal violet was added for staining for 5 min. The plates were then washed with sterile normal saline solution three times. Finally, the stain was resolubilized in 95% ethanol and absorbance was measured at 600 nm after 30 min.

### 4.11. In Vivo Acute Toxicity

Male C57BL/6 mice (6–8 weeks of age) were randomly divided into three groups of eight mice per group. Either normal saline or RK22 (20 and 40 mg/kg) was injected intravenously via the tail vein. Mice were observed for survival status within 96 h of the injection and blood was collected via retro-orbital bleeding for routine blood analysis. For histopathological analysis, sections of lung, liver, spleen, and kidney were fixed with 4% paraformaldehyde, embedded in paraffin wax, and stained with hematoxylin and eosin (H&E).

### 4.12. S. aureus-Induced Bacteremia Model

*S. aureus* ATCC6538 or methicillin-resistant *S. aureus* MRSA-Z were diluted to a concentration of 1 × 10^9^ CFU/mL with normal saline. Male C57BL/6 mice (6–8 weeks of age) were randomly divided into seven groups of 10–12 mice per group. *S. aureus* ATCC6538 (100 μL) or methicillin-resistant *S. aureus* MRSA-Z (200 μL) were injected intravenously. At the end of 1 h, mice received an intraperitoneal injection of normal saline, vancomycin (dissolved in saline with the concentration of 2 and 4 mg/kg), or RK22 (dissolved in saline with the concentration of 2, 4, and 8 mg/kg). At 4 h post-administration, the mice were sacrificed, and blood and lung samples were collected. Lungs were then weighed and placed in 1 mL of PBS and homogenized on ice with a tissue grinder. Homogenates and clarified blood were plated in LB agar culture medium and CFUs were counted after 24 h. For histopathological analysis, sections of lung were fixed with 4% paraformaldehyde, embedded in paraffin wax, and stained with H&E.

### 4.13. Enzymatic Activity Assay of Coagulation Factors

The enzymatic activity of RK22 and LL-37 against coagulation factors was measured using the chromogenic substrate method [16]. In brief, 5 μL of FXa (Enzymeresearch, Lake Oswego, OR, USA) (final concentration 3 μg/mL) or thrombin (Enzymeresearch, Lake Oswego, OR, USA) (final concentration 0.1056 μg/mL) and 5 μL of sample (final concentration 40, 200, and 1000 nM) were incubated for 10 min in 96-well plates, respectively. Following incubation, thrombin substrate (S-2238, Chromogenix, Milano, Italy, 45 μg/mL) or FXa substrate (S-2222, Chromogenix, Milano, Italy, 90 μg/mL) dissolved with 90 μL of enzyme dynamic buffer (100 mM NaCl, 50 mM Tris, 5 mM CaCl_2_, pH 8.0) were added and absorbance was measured at 405 nm every 1 min for 90 min.

### 4.14. Platelet Activity Assay

Platelets were isolated from the mice as described in a previous study with minor modifications [24]. The effect of RK22 and LL-37 on platelet activity was detected by flow cytometry. Mouse blood was mixed with Tyrode’s buffer A (137 mM NaCl, 2 mM KCl, 0.3 mM NaH_2_PO_4_, 12 mM NaHCO_3_, 5.5 mM glucose, 0.35% BSA, 1 mM MgCl_2_, and 0.2 mM EDTA, pH 6.5) and centrifuged (150× *g*, 10 min, room temperature) to obtain platelet-rich plasma (PRP), which was then pelleted (400× *g*, 5 min, room temperature), washed with Tyrode’s buffer A, and finally resuspended in Tyrode’s buffer B (137 mM NaCl, 2 mM KCl, 0.3 mM NaH_2_PO_4_, 12 mM NaHCO_3_, 5.5 mM glucose, 0.35% BSA and 2 mM CaCl_2_, pH 7.4). Antibodies against P-selectin (Biolegend, San Diego, CA, USA) were added to the platelets, followed by incubation for 10 min at room temperature. Samples (20, 10, 5, and 2.5 μg/mL) or normal saline were added, respectively, and analyzed by flow cytometry (BD, LSRFortessa^TM^, Franklin Lakes, NJ, USA).

### 4.15. Statistical Analysis

The data obtained from independent experiments are presented as mean ± standard deviation (SD). For normal continuous variables, one-way analysis of variance (ANOVA) was used. All analyses were conducted using GraphPad Prism v5. Asterisks represent *p*-value classifications: * *p* < 0.05; ** *p* < 0.01, and *** *p* < 0.001.

## 5. Conclusions

In conclusion, we identified RK22 as a novel and highly effective antimicrobial peptide against *S. aureus* and MRSA, with good plasma stability, negligible cytotoxicity, minimal hemolytic activity, and no significant promotion of the coagulation system. These findings establish RK22 as an excellent candidate or template for the development of therapeutic agents aimed at treating *S. aureus* and MRSA infections.

## Figures and Tables

**Figure 1 ijms-24-13453-f001:**
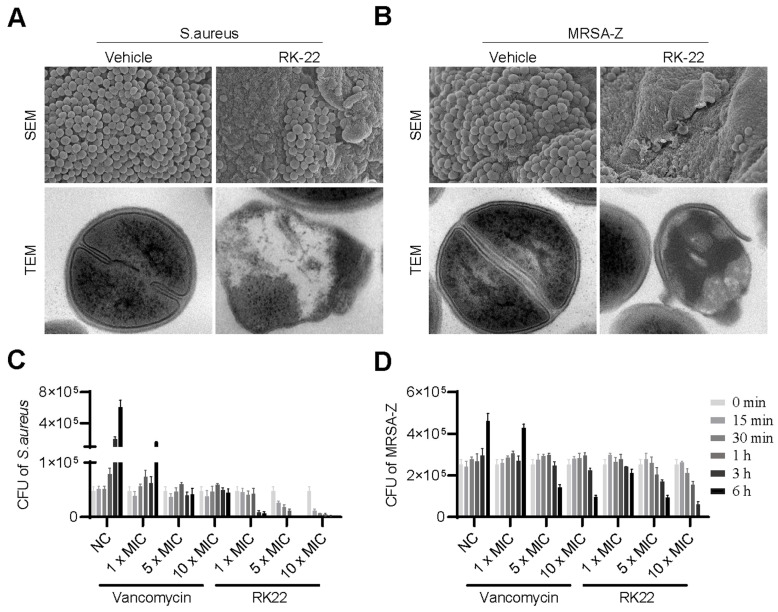
RK22 kills *S. aureus* ATCC6538 and MRSA-Z by permeabilizing the cell membrane. After treatment with or without RK22 (100 μg/mL) for 4 h, the membrane morphology of *S. aureus* ATCC6538 (**A**) and MRSA-Z (**B**) was determined by SEM and TEM. Killing kinetics of RK22 against *S. aureus* ATCC6538 (**C**) and MRSA-Z (**D**) were investigated. Data represent means ± SD of 3 individual experiments.

**Figure 2 ijms-24-13453-f002:**
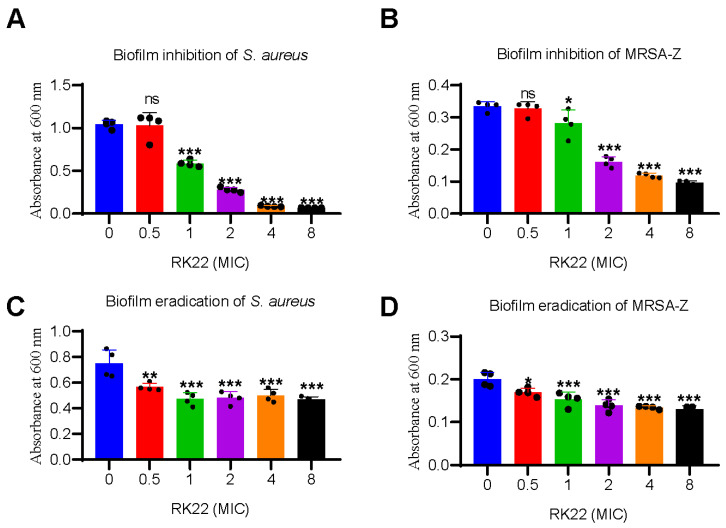
RK22 exhibits biofilm inhibition and eradication activities. Biofilm was measured at 600 nm after staining with 0.1% crystal violet. Inhibitory effects of RK22 on the biofilm formation of *S. aureus* ATCC6538 (**A**) and MRSA-Z (**B**) were analyzed after incubation with RK22 (final concentration 0.5, 1, 2, 4, 8 × MIC) with the bacteria for 24 h. Furthermore, to determine the effects of RK22 on the biofilm eradication, RK22 (final concentration 0.5, 1, 2, 4, 8 × MIC) was added into the wells with an established biofilm of *S. aureus* ATCC6538 (**C**) and MRSA-Z (**D**). Data represent means ± SD of 3 individual experiments. ns: difference was not significant. * *p* < 0.05, ** *p* < 0.01, *** *p* < 0.001.

**Figure 3 ijms-24-13453-f003:**
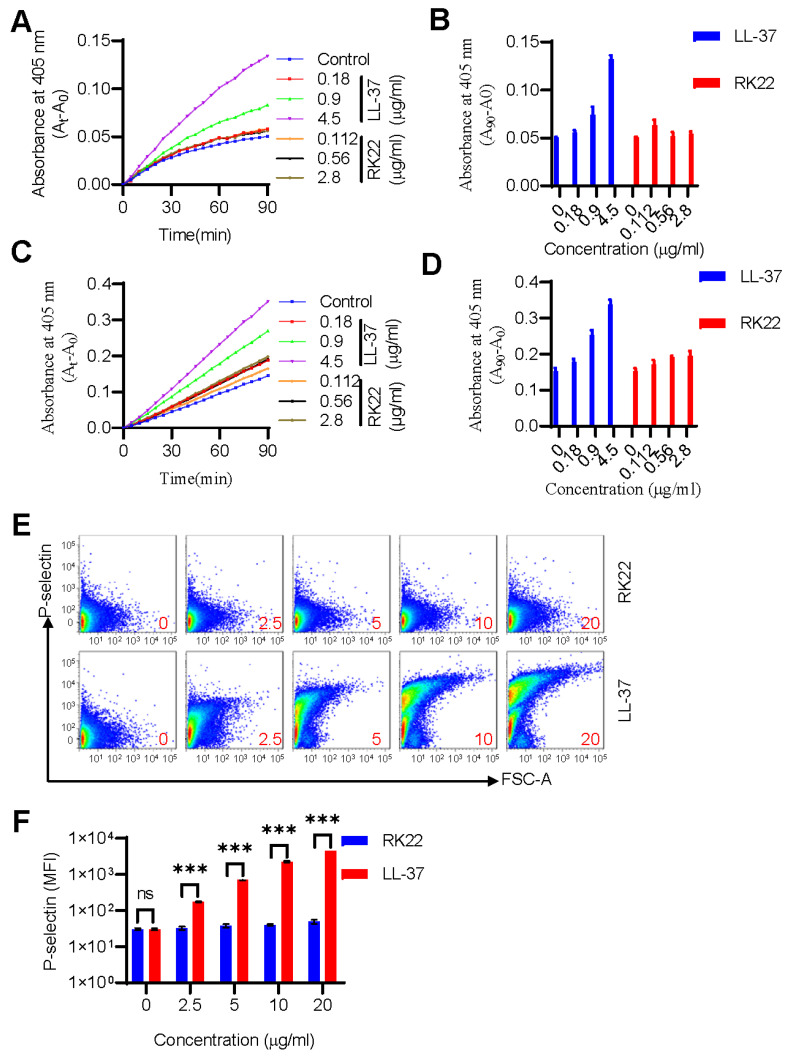
Effects of RK22 on the enzymatic activity of coagulation factors and platelet activation. Compared with the enhancement of LL-37 (final concentration 40 nM (0.18 μg/mL), 200 nM (0.9 μg/mL), 1000 nM (4.5 μg/mL)) on the enzymatic activity of thrombin and FXa, RK22 (final concentration 40 nM (0.112 μg/mL), 200 nM (0.56 μg/mL), 1000 nM (2.8 μg/mL)) showed no significant promotion on thrombin (**A**,**B**) and FXa (**C**,**D**). Furthermore, RK22 or LL-37 (20, 10, 5, 2.5 μg/mL) was incubated with washed platelets for 10 min, and then platelet activation was determined and analyzed by flow cytometry (**E**,**F**). Data represent means ± SD of 3 individual experiments. ns: difference was not significant. *** *p* < 0.001.

**Figure 4 ijms-24-13453-f004:**
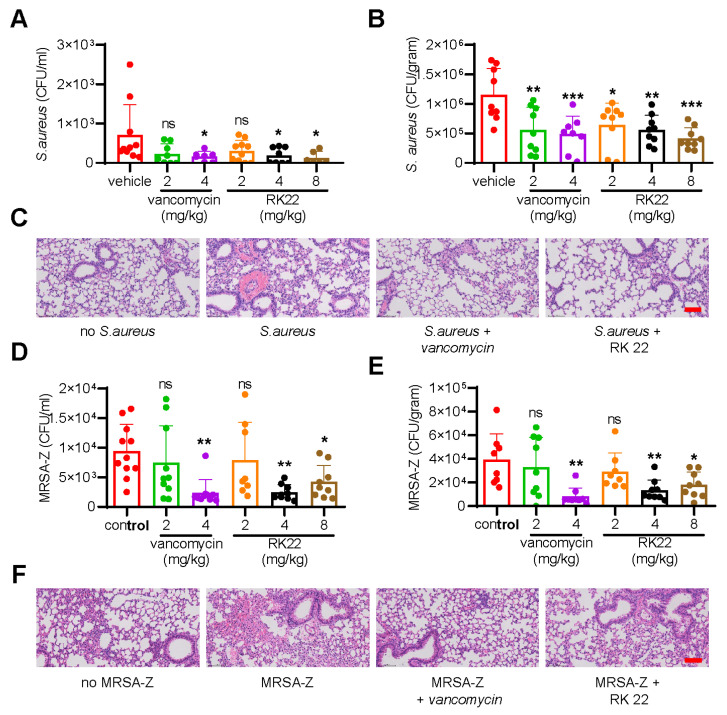
RK22 suppresses the dissemination of *S. aureus* (ATCC6538) and MRSA-Z in vivo. Mice (n = 10–12 mice per group) were injected with *S. aureus* (ATCC6538 1 × 10^8^ CFU/mouse, ip or MRSA-Z, 2 × 10^8^ CFU/mouse, ip); 1 h later, samples (RK22: 2, 4, 8 mg/kg; vancomycin: 2, 4 mg/kg, saline as the negative control group) were intraperitoneally injected with the concentration indicated. After 4 h of administration, mice were sacrificed, blood (**A**,**D**) and lung (**B**,**E**) were collected for the measurement of bacterial loads. Additionally, the sections of lung were stained with hematoxylin & eosin (H&E) for histopathological analysis (**C**,**F**), scale bar: 100 μm. Data represent mean ± SD values of six independent experiments. * *p* < 0.05, ** *p* < 0.01, *** *p* < 0.001.

**Table 1 ijms-24-13453-t001:** Physicochemical properties of peptides.

Peptide	Sequence	L	NC	H	PR (n/%)	NPR (n/%)
KS14	KSSNTKAKKKKKNN	14	+7	−0.589	13/92.86	1/7.14
RA19	RAVLCPKPKPKKKKVCVVL	19	+7	0.362	7/36.84	12/63.16
KT20	KTRRRNRKHKKTINTETVQI	20	+7	−0.203	17/85	3/15
VF21	VFLICFSLISPASFENVRAKW	21	+1	0.790	7/33.33	14/66.67
RK22	RKYKEKKDKSQNKKKKRKCMIL	22	+10	−0.316	17/77.27	5/22.73
KS34	KSKSKKPSKSKPKKKKTSVVQQELQDVISFDFGC	34	+7	0.029	24/70.59	10/29.41

L: length; NC: Net charge; H: Hydrophobicity; PR: Polar residues; NPR: Nonpolar residues.

**Table 2 ijms-24-13453-t002:** Prediction score of peptides.

Peptide	Random Forest Class	Earth Class	Earth Prob. Source	*E. coli* Prob.	*S. aureus* Prob.
KS14	Non-AMP	AMP	0.52	0.63	0.37
RA19	AMP	Non-AMP	0.21	0.93	0.07
KT20	AMP	Non-AMP	0.02	0.74	0.26
VF21	AMP	AMP	0.66	0.3	0.7
RK22	AMP	AMP	0.5	0.52	0.48
KS34	AMP	Non-AMP	0.15	0.86	0.14

Prob.: probability.

**Table 3 ijms-24-13453-t003:** Antimicrobial activity of the peptides against four standard strains.

Bacteria Strain	MIC (μg/mL)					
	KS14	KS34	KT20	RK22	VF21	RA19
*S. aureus* (ATCC6538)	50	>100	50	6.25	>100	12.5
*A. baumannii*, ATCC19606	>100	>100	>100	100	>100	>100
*P. aeruginosa*, ATCC9027	>100	>100	>100	>100	>100	>100
*E. coli*, ATCC25922	>100	>100	>100	25	>100	>100

**Table 4 ijms-24-13453-t004:** Antimicrobial activity of RK22 against *S. aureus* and five clinical isolates of *S. aureus*.

Bacteria Strain	MIC (μg/mL)	
	RK22	Vancomycin
*S. aureus* (ATCC6538)	6.25	1.56
MRSA-Z	6.25	1.56
MRSA11	6.25	1.56
MRSA22	6.25	1.56
SA220823	12.5	1.56
SA15772	12.5	1.56

## Data Availability

For original data, please contact duanzilei@mail.kiz.ac.cn (Z.D.) and rlai@mail.kiz.ac.cn (R.L.).

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
