# Peer review of "Identification and Characterization of RK22, a Novel Antimicrobial Peptide from Hirudinaria manillensis against Methicillin Resistant Staphylococcus aureus"

_ijms, 2023, doi:10.3390/ijms241713453_

Round 1

Reviewer 1 Report

Authors report the identification of a new antimicrobial peptide (RK22) from the salivary gland transcriptome of Hirudinaria manillensis. This peptide seems to have good antimicrobial activity against S. aureus, and the inhibitory capacity of biofilm formation. Furthermore, the authors report that RK22 possesses good plasma stability, negligible cytotoxicity, minimal hemolytic activity, and no significant influence on the coagulation system. Authors report the results about a great work, but the introduction section, the results presentation  and the discussion section need to be improved

Furthermore, I would suggest not to show the antimicrobial activity of RK22  against other bacteria, but to immediately focus on S. aureus. The cytotoxic activity of this peptide appears to be present at concentrations higher than even the concentration used as antibacterial against E. coli. Therefore, if authors prefer to keep the antibacterial activity data against the other bacterial strains, I would still suggest discussing this in the discussion section.

Title: I would suggest changing the title to make it clear that authors reported of peptides synthesized after in silico analysis

Abstract: also in this section it should be clarified that these are synthetic peptides after bioinformatic analysis

Introducion: the entire introduction section should be improved by adding the state of the art on Hirudinaria manillensis salivary gland properties.

Lane 76: authors report references 35-43, but in the references I find up to 41: is it a typo? It should be re-checked

Lanes 77-81: this sentence refers to the results. Please, move it to the results section. Here, limit yourself to describing the scope of your work

Results:  I suggest to move table S1 in the main text. Moreover, do not present tables before having written the results. Please, move tables after the text where are cited.

Check that S. aureus is italicized throughout the text

Lane 88: Please, delete  "(https://www.dveltri.com/as- can/v2/index.html).". This must be reported in material and methods section

Table 1: what do you mean with Prob.? Please, clarify

Table 2: It's necessary "Bacteria strain" and not "Bacteria strains". Please, write in the same mode the bacterial name 

Figure 1: I suggest to replace 1xMIC etc..in the x axis  with the concentrations used  (ug/ml) of both vancomycin and RK22

Figure 2: I suggest to add the strain name in the  histogram titles to better understand the data

Table 4: In my opinion, it's a superfluous table. Just report the data in the text since there are no differences over time

Figure 3: I suggest to divide  data regarding LL37 from that regarding RK22, in graphs A and C, in order to better note differences.

Please, uniforms the measure units in all the text: nM or ug/ml

Figure 4:  (A) Maybe it should have been CFU/ml? Please check. Also in this case I suggest to add titles in each graph to immediately understand which strain it is.

Discussion:

In my opinion, the discussion should be improved by commenting and contextualizing all the results obtained

Material and methods:

lanes 306: how were the peptides resuspended?

Lane 320: why did you choose HEK293T? Why weren't cytotoxicity tests also performed on PBMCs?

Lane 326: Please, specify the company of the CCK8 used

Lane 329: Please, add the reference relative this paragraph

Please, specify the company of all reagents were used

Author Response

Point 1. Authors report the identification of a new antimicrobial peptide (RK22) from the salivary gland transcriptome of Hirudinaria manillensis. This peptide seems to have good antimicrobial activity against S. aureus, and the inhibitory capacity of biofilm formation. Furthermore, the authors report that RK22 possesses good plasma stability, negligible cytotoxicity, minimal hemolytic activity, and no significant influence on the coagulation system. Authors report the results about a great work, but the introduction section, the results presentation and the discussion section need to be improved.

Response 1: Many thanks for your review. We have corrected it and improved the introduction section, the results presentation and the discussion section in the revised manuscript.

Point 2. Furthermore, I would suggest not to show the antimicrobial activity of RK22 against other bacteria, but to immediately focus on S. aureus. The cytotoxic activity of this peptide appears to be present at concentrations higher than even the concentration used as antibacterial against E. coli. Therefore, if authors prefer to keep the antibacterial activity data against the other bacterial strains, I would still suggest discussing this in the discussion section.

Response 2: Many thanks for your review and suggestions. We have decided to keep the antibacterial data against the other bacterial strains, which has been discussed in the discussion section in the revised manuscript. Furthermore, RK22 exhibited potent antibacterial activity against S. aureus ATCC6538, A. baumannii ATCC19606, and E. coli ATCC8739, with MIC values of 6.25, 100, and 25 μg/mL, respectively, and RK22 showed no cytotoxic activity even at the concentration of 400 μg/mL, which suggested the safety of RK22.

Point 3. Title: I would suggest changing the title to make it clear that authors reported of peptides synthesized after in silico analysis

Response 3: Many thanks for your review and suggestion. We have clarified that these are synthetic peptides in the abstract and methods section, so we decided not to change the title.

Point 4. Abstract: also in this section it should be clarified that these are synthetic peptides after bioinformatic analysis.

Response 4: Many thanks for your review and suggestion. We have clarified that RK22 is the synthetic peptide in abstract of the revised manuscript.

Point 5. Introducion: the entire introduction section should be improved by adding the state of the art on Hirudinaria manillensis salivary gland properties.

Response 5: Many thanks for your review and suggestion. We have reorganized and described the introduction section with the addition of the state of the art on Hirudinaria manillensis salivary gland properties in the revised manuscript.

Point 6. Lane 76: authors report references 35-43, but in the references I find up to 41: is it a typo? It should be re-checked

Response 6: Many thanks for your review. We have checked and corrected it in the revised manuscript.

Point 7. Lanes 77-81: this sentence refers to the results. Please, move it to the results section. Here, limit yourself to describing the scope of your work.

Response 7: Many thanks for your review and suggestion. We have checked and corrected it in the revised manuscript.

Point 8. Results: I suggest to move table S1 in the main text. Moreover, do not present tables before having written the results. Please, move tables after the text where are cited.

Response 8: Many thanks for your review and suggestion. We have moved table S1 in the main test in the revised manuscript. Furthermore, we have moved tables and figures after having written the results.

Point 9. Check that S. aureus is italicized throughout the text

Response 9: Many thanks for your review. We have checked and corrected it in the revised manuscript.

Point 10. Lane 88: Please, delete "(https://www.dveltri.com/as- can/v2/index.html).". This must be reported in material and methods section

Response 10: Many thanks for your review. We have deleted it in the revised manuscript.

Point 11. Table 1: what do you mean with Prob.? Please, clarify

Response 11: Many thanks for your review. Prob. means probability, and we have clarified it in the revised manuscript.

Point 12. Table 2: It's necessary "Bacteria strain" and not "Bacteria strains". Please, write in the same mode the bacterial name

Response 12: Many thanks for your review and suggestion. We have corrected it in the revised manuscript.

Point 13. Figure 1: I suggest to replace 1xMIC etc..in the x axis  with the concentrations used  (ug/ml) of both vancomycin and RK22

Response 13: Many thanks for your review and suggestion. This description using 1x MIC etc is acceptable (Mwangi, J.; Yin, Y.; Wang, G.; Yang, M.; Li, Y.; Zhang, Z.; Lai, R., The antimicrobial peptide ZY4 combats multidrug-resistant Pseudomonas aeruginosa and Acinetobacter baumannii infection. Proceedings of the National Academy of Sciences of the United States of America 2019, 116, (52), 26516-26522.), and we did not changed it in the revised manuscript.

Point 14. Figure 2: I suggest to add the strain name in the histogram titles to better understand the data

Response 14: Many thanks for your review and suggestion. We have added the strain name in the revised manuscript.

Point 15. Table 4: In my opinion, it's a superfluous table. Just report the data in the text since there are no differences over time

Response 15: Many thanks for your review. Since the antibacterial ability of RK22 has not changed, it may appear that this table is superfluous, but this data is necessary

Point 16. Figure 3: I suggest to divide data regarding LL37 from that regarding RK22, in graphs A and C, in order to better note differences.

Response 16: Many thanks for your review and suggestion. Figure A and C indicated the representative curve of each concentration, and Figure B and D showed the better note difference.

Point 17. Please, uniforms the measure units in all the text: nM or ug/ml

Response 17: Many thanks for your review. We have corrected it in the revised manuscript.

Point 18. Figure 4:  (A) Maybe it should have been CFU/ml? Please check. Also in this case I suggest to add titles in each graph to immediately understand which strain it is.

Response 18: Many thanks for your review. Indeed, it should have been CFU/ml in the figure A and we have corrected it in the revised manuscript. Furthermore, we have added the bacterial strain in each graph in the revised manuscript.

Point 19. Discussion:

In my opinion, the discussion should be improved by commenting and contextualizing all the results obtained

Response 19: Many thanks for your review and suggestion. We have improved the discussion section in the revised manuscript.

Point 20. lanes 306: how were the peptides resuspended?

Response 20: Many thanks for your review. The peptides were resuspended using RPMI 1640 medium for the concentration of 0–200 μg/mL, which was clarified in the revised manuscript.

Point 21.  Lane 320: why did you choose HEK293T? Why weren't cytotoxicity tests also performed on PBMCs?

Response 21: Many thanks for your review. HEK293T cell is one of the normal cell line, which is commonly used in the cytotoxicity experiment. For the isolation of PBMCs, we need to find volunteers and obtain the approval from the the Ethics Committee, and the injection of RK22 (20 and 40 mg/kg) intravenously via the tail vein showed no acute toxicity in vivo, so we we did not performed the cytotoxicity tests using PBMCs.

Point 22.  Lane 326: Please, specify the company of the CCK8 used

Response 22: Many thanks for your review. We have added the company of the CCK8 used in the revised manuscript.

Point 23.  Lane 329: Please, add the reference relative this paragraph

Response 23: Many thanks for your review. we have added the reference relative this paragraph in the revised manuscript.

Point 24.  Please, specify the company of all reagents were used

Response 24: Many thanks for your review. we have specified the company of all reagents were used in the revised manuscript.

Reviewer 2 Report

The article describe the discovery of a novel peptide with antibacterial properties against S. aureus, including antibiotic-resistant strains. The peptide was also tested for its antibacterial properties in vivo, as well as possible side effects where it was found not to have acute toxicity or thrombotic properties etc. The topic is extremely important due to the reduced number of available antibiotics in the market against resistant bacterial strains.

I recommend the publication of the article in the IJMS after minor corrections:

a) more information about the organism from where the peptide was isolated should be given in the introduction

b) more information about the structure of the peptide should be given in the introduction. Is there any structure-activity relationship?

c) More discussion about the cause of the biological activity against S. aureus and its selectivity, should be given in the discussion

d) line 76: references 35-43 : There are only 41 references.

e) the fixation of the bacteria before TEM images should be described or references should be given

f) line 144: Section number should be 2.2 and corrections of the following section numbers

Author Response

Point 1. more information about the organism from where the peptide was isolated should be given in the introduction.

Response 1: Many thanks for your review and suggestion, we have added the information of the H. manillensis in the introduction section in the revised manuscript.

Point 2. more information about the structure of the peptide should be given in the introduction. Is there any structure-activity relationship?

Response 2: Many thanks for your review and suggestion. Peptides studied in this paper are positive charged peptides as added in the introduction section in the revised manuscript, which might determine the antimicrobial activity. Furthermore, there must be the structure-activity relationship, because only RK22 showed the excellent antimicrobial activity with all the peptides rich in the positive-charged amino acids. However, we did not investigate the structure-activity relationship in this paper, and we only predicted the probability through bioinformatic analysis. Further research on the  structure-activity relationship needed to be investigated in the future.

Point 3. More discussion about the cause of the biological activity against S. aureus and its selectivity, should be given in the discussion.

Response 3: Many thanks for your review. We have added the discussion in the revised manuscript.

Point 4. line 76: references 35-43 : There are only 41 references.

Response 4: Many thanks for your review. We have checked and corrected it in the revised manuscript.

Point 5. the fixation of the bacteria before TEM images should be described or references should be given

Response 5: Many thanks for your review and suggestion. We have described and corrected it in the revised manuscript.

Point 6. line 144: Section number should be 2.2 and corrections of the following section numbers

Response 6: Many thanks for your review. In the front of this section, there is 2.2 and 2.3, so it is correct with the section number 2.4 here.

Round 2

Reviewer 1 Report

 Title: I suggest to change the title in "Identification and characterization of RK22, a novel antimicrobial peptide from Hirudinaria manillensis against methicillin resistant Staphylococcus aureus" 

lane 76: please, delete "successively"

Table1, 2 and 3: in the title delete "these"

lane 108: "strains" is correct 

lane 116: title Table 3 "Antimicrobial activity of the peptides against four standard strains" and not " Antimicrobial activity of these peptides against four standard strain". Moreover, "Bacteria strain" and not "bacteria strain". 

Table 4: "Bacteria strain" and not "bacteria strain". S. aureus in italic

Figura 1: In A and B specify the RK22 concentration used

Table 5: I still think this table should be deleted. It does not give additional information to what is said in the text.

Figure 4 C. please, add the magnification used for the microscope observation and an arrow to highlight the presence of S. aureus.

lane 410: "2.5% glutaraldehyde (China)" maybe something is missing?

Authors reported that "Peptides were resuspended in RPMI 1640 medium": also when injected in mice?

Author Response

Point 1. Title: I suggest to change the title in "Identification and characterization of RK22, a novel antimicrobial peptide from Hirudinaria manillensis against methicillin resistant Staphylococcus aureus"

Response 1: Many thanks for your suggestion. We have corrected it in the revised manuscript.

Point 2. lane 76: please, delete "successively"

Response 2: Many thanks for your review and suggestion. We have deleted "successively" in the revised manuscript.

Point 3. Table1, 2 and 3: in the title delete "these"

Response 3: Many thanks for your review and suggestion. We have deleted " these " in the revised manuscript.

Point 4. lane 108: "strains" is correct

Response 4: Many thanks for your review and suggestion. We have corrected it in the revised manuscript.

Point 5. lane 116: title Table 3 "Antimicrobial activity of the peptides against four standard strains" and not " Antimicrobial activity of these peptides against four standard strain". Moreover, "Bacteria strain" and not "bacteria strain".

Response 5: Many thanks for your review and suggestion. We have corrected them in the revised manuscript.

Point 6. Table 4: "Bacteria strain" and not "bacteria strain". S. aureus in italic

Response 6: Many thanks for your review. We have corrected them in the revised manuscript.

Point 7. Figura 1: In A and B specify the RK22 concentration used

Response 7: Many thanks for your review. We have added the concentration of RK22 it in the revised manuscript.

Point 8. Table 5: I still think this table should be deleted. It does not give additional information to what is said in the text.

Response 8: Many thanks for your review and suggestion. We have moved table 5 to the Supplementary materials as the Table S1.

Point 9. Figure 4 C. please, add the magnification used for the microscope observation and an arrow to highlight the presence of S. aureus.

Response 9: Many thanks for your review. The scale bar is 100 μm indicated in the figure legends, and the diameter of S.aureus is about 1 μm, so we can not detect and point out the S.aureus.  

Point 10. lane 410: "2.5% glutaraldehyde (China)" maybe something is missing?

Response 10: Many thanks for your review. We have added the company of the reagent in the revised manuscript.

Point 11. Authors reported that "Peptides were resuspended in RPMI 1640 medium": also when injected in mice?

Response 11: Many thanks for your review. When injected in mice, peptides were resuspended in saline, not the RPMI 1640 medium, which was pointed out in the revised manuscript.